# The Landscape of Transmembrane Protein Family Members in Head and Neck Cancers: Their Biological Role and Diagnostic Utility

**DOI:** 10.3390/cancers13194737

**Published:** 2021-09-22

**Authors:** Oliwia Koteluk, Antonina Bielicka, Żaneta Lemańska, Kacper Jóźwiak, Weronika Klawiter, Andrzej Mackiewicz, Urszula Kazimierczak, Tomasz Kolenda

**Affiliations:** 1Department of Cancer Immunology, Chair of Medical Biotechnology, Poznan University of Medical Sciences, 8 Rokietnicka Street, 60-806 Poznan, Poland; zazet13@gmail.com (Ż.L.); kacperjozwiak97@gmail.com (K.J.); klawweronika@gmail.com (W.K.); andrzej.mackiewicz@wco.pl (A.M.); ukazimierczak@gmail.com (U.K.); kolenda.tomek@gmail.com (T.K.); 2Department of Diagnostics and Cancer Immunology, Greater Poland Cancer Centre, 15 Garbary Street, 61-866 Poznan, Poland; 3Laboratory of Cancer Genetics, Greater Poland Cancer Centre, 15 Garbary Street, 61-866 Poznan, Poland

**Keywords:** TMEM, HNSCC, biomarker, immune response, TCGA, HPV

## Abstract

**Simple Summary:**

Transmembrane proteins (TMEM) are a large group of integral membrane proteins whose molecular and biological functions are not fully understood. It is known that some of them are involved in tumor formation and metastasis. Here, we performed a panel of TCGA data analyses to investigate the role of different *TMEM* genes in head and neck squamous cell carcinoma (HNSCC) and define their potential as biomarkers. Based on changes in the expression levels in HNSCC tumors, we selected four *TMEM* genes: *ANO1*, *TMEM156*, *TMEM173*, and *TMEM213* and associated them with patient survival. We also demonstrated that the expression of those *TMEMs* highly correlates with the enrichment of genes involved in numerous biological processes, especially metastasis formation and immune response. Thus, we propose *ANO1*, *TMEM156*, *TMEM173*, and *TMEM213* as new biomarkers and potential targets for personalized therapy of HNSCC.

**Abstract:**

*Background:* Transmembrane proteins (TMEM) constitute a large family of proteins spanning the entirety of the lipid bilayer. However, there is still a lack of knowledge about their function or mechanism of action. In this study, we analyzed the expression of selected *TMEM* genes in patients with head and neck squamous cell carcinoma (HNSCC) to learn their role in tumor formation and metastasis. *Materials and Methods:* Using TCGA data, we analyzed the expression levels of different *TMEMs* in both normal and tumor samples and compared those two groups depending on clinical-pathological parameters. We selected four TMEMs whose expression was highly correlated with patient survival status and subjected them to further analysis. The pathway analysis using REACTOME and the gene set enrichment analysis (GSEA) were performed to evaluate the association of those *TMEMs* with genes involved in hallmarks of cancer as well as in oncogenic and immune-related pathways. In addition, the fractions of different immune cell subpopulations depending on *TMEM* expression were estimated in analyzed patients. The results for selected *TMEMs* were validated using GEO data. All analyses were performed using the R package, Statistica, and Graphpad Prism. *Results:* We demonstrated that 73% of the analyzed *TMEMs* were dysregulated in HNSCC and depended on tumor localization, smoking, alcohol consumption, or HPV infection. The expression levels of *ANO1*, *TMEM156*, *TMEM173*, and *TMEM213* correlated with patient survival. The four *TMEMs* were also upregulated in HPV-positive patients. The elevated expression of those *TMEMs* correlated with the enrichment of genes involved in cancer-related processes, including immune response. Specifically, overexpression of *TMEM156* and *TMEM173* was associated with immune cell mobilization and better survival rates, while the elevated *ANO1* expression was linked with metastasis formation and worse survival. *Conclusions:* In this work, we performed a panel of in silico analyses to discover the role of *TMEMs* in head and neck squamous cell carcinoma. We found that *ANO1*, *TMEM156*, *TMEM173*, and *TMEM213* correlated with clinical status and immune responses in HNSCC patients, pointing them as biomarkers for a better prognosis and treatment. This is the first study describing such the role of *TMEMs* in HNSCC. Future clinical trials should confirm the potential of those genes as targets for personalized therapy of HNSCC.

## 1. Introduction

Head and neck squamous cell carcinomas (HNSCCs) are aggressive malignancies with high morbidity and mortality. Worldwide, HNSCCs are responsible for over 550,000 new cases and over 380,000 deaths per year [1]. The main risk factors of HNSCCs are long-term tobacco use, alcohol consumption, and infection with high-risk types of human papillomavirus (HPV). HNSCCs arise from stratified epithelial cells and can be located in the oral cavity, pharynx, or larynx. Treatment strategies of HNSCCs include surgery, irradiation, and platinum-based chemotherapy. However, existing therapies are not sufficient, and the risk of relapse is still high. Moreover, treatments such as radio- or chemotherapy are associated with toxicity to other organs and can lead to the reduction of the quality of life. Thus, there is a constant need for better therapeutic strategies. The most promising option remains the targeted therapy [2,3,4]. There is only one such therapy for HNSCC approved by the FDA- cetuximab, a monoclonal antibody that targets the epidermal growth factor receptor (EGFR) [5]. To improve HNSCC treatment, personalized therapy should be applied as the first matter. For that purpose, there is an urgent need to develop specific biomarkers.

The transmembrane protein (TMEM) family is a large group of proteins that span the lipid bilayer. Their structure, biological function, and mechanism of action are poorly understood, mainly due to difficulties in their extraction and purification [5,6]. TMEMs can be found in various cell types and cellular membranes, including mitochondria, endoplasmic reticulum (ER), lysosomes, or Golgi apparatus. Several studies have shown that different TMEM genes can be up or downregulated in cancers and may act as tumor suppressors or oncogenes. Their role has also been described in chemoresistance and response to anticancer therapy [6]. Furthermore, lots of TMEM proteins are known to be involved in cancer-related signaling pathways like EGFR-induced NF-κB activation or TGF-β signaling [6,7]. Several TMEMs meet the criteria of prognostic biomarkers, as their expression has been correlated with metastasis, tumor recurrence, and patient survival [5,8,9]. Identifying TMEMs engaged in tumor development and progression is a promising approach to finding the new therapeutic targets for cancer treatment.

In this study, we examined twenty-two different TMEM genes in patients with HNSCC and analyzed their connection with clinical attributes, immune response, and hallmarks of cancer. Using a panel of bioinformatics tools, we performed in silico analyses of the expression data collected within The Cancer Genome Atlas Project (TCGA). We selected four *TMEMs*-*ANO1*, *TMEM156*, *TMEM173*, and *TMEM213* as potential biomarkers for a better diagnosis and treatment of HNSCC.

## 2. Materials and Methods

### 2.1. Data Collection

The expression profiles of 22 TMEM genes (ANO1, TMEM17, TMEM25, TMEM45A, TMEM45B, TMEM48, TMEM88, TMEM97, TMEM98, TMEM140, TMEM156, TMEM158, TMEM173, TMEM176A, TMEM206, RTP3, TMEM22, TMEM30B, TMEM43, TMEM61, TMEM116, TMEM213), clinical and pathological data of 522 HNSCC patients, and 44 adjacent normal tissue samples were downloaded from TCGA database (TCGA Head and Neck Cancer; dataset ID: TCGA.HNSC.sampleMap/HiSeqV2_PANCAN; pan-cancer normalized log2(norm_count + 1) and dataset ID: TCGA.HNSC.sampleMap/HNSC_clinicalMatrix) using USCS Xena Browser [10], and UALCAN [11] database (http://ualcan.path.uab.edu/index.html; level 3 TCGA RNA-seq data, accessed on 18 November 2020). The list of genes correlated with TMEMs was downloaded from cBioPortal [12,13].

### 2.2. Clinical and Pathological Data Analysis

The expression levels of selected *TMEMs* in normal tissues vs. primary tumors were downloaded from the UALCAN database and visualized on graphs as transcripts per million (TPM). Differences between the samples were assessed as described previously [11], using t-test and PERL script with comprehensive perl archive network (CPAN) module. Statistically significant data were taken for further analysis. The correlation between the expression levels of selected genes was measured and the heatmap presenting the R-coefficient values of each correlation was created using the Morpheus online tool (https://software.broadinstitute.org/morpheus, accessed on 25 November 2020). Next, the receiver operating characteristic curve (ROC) analysis of *TMEMs* was performed and the area under the curve (AUC) comparing paired adjacent normal tissues was estimated as described in Section 2.5. All 522 patients were divided into three groups depending on tumor localization (oral cavity, pharynx, or larynx), based on National Institute of Health guidelines [14] and the expression of *TMEM*s was analyzed as described in Section 2.5.

The following clinical-pathological parameters were analyzed: age (<61 vs. ≥61), gender (female vs. male), alcohol consumption (yes vs. no), smoking (no/ex vs. yes), stage (I + II vs. III + IV), T stage (T1 + T2 vs. T3 + T4), N-stage (N0 vs. N1 + N2 + N3), tumor grade (G1 + G2 vs. G3 + G4), angiolymphatic invasion (positive vs. negative), lymph node dissection (yes vs. no), and HPV status (positive vs. negative) as described in Section 2.5. Next, patients were divided into two groups with high or low expression levels of specific *TMEMs* (based on the expression mean) and overall survival (OS) and disease-free survival (DFS) analyses were performed with 3-year follow-up as described in Section 2.5. Patients’ characteristics are included in the Appendix A.

### 2.3. Pathway and Gene Set Enrichment Analysis

The list of genes correlated with selected *TMEMs* were downloaded from cBioPortal [12,13]. For further analysis only genes with Spearman’s correlation coefficient R > 0.3 (for positive correlation) and R < −0.3 (for negative correlation) were taken. The R software version 4.0.3. (R Core Team, Vienna, Austria) [15], ReactomePA [16], and org.Hs.eg.db [17] packages were applied to perform pathway enrichment analysis. Fifteen statistically significant (*p* < 0.05) pathways with the highest gene count were visualized using the “ggplot2” package [18]. For gene set enrichment analysis (GSEA) (v4.1.0) [19] patients were divided into two groups based on the mean expression of *ANO1*, *TMEM156*, *TMEM173*, or *TMEM213* and hallmarks (H), oncogenic (C6), and immunologic signature gene sets (C7) were analyzed [20]. The ranked list of genes consisted of 20,530 genes with the permutation number set at 1000. False discovery rate (FDR) ≤ 0.25 and *p*-value < 0.05 were considered significant as described previously [21].

### 2.4. Analysis of Stromal and Immune Cell Fractions

Immune, stromal, and ESTIMATE scores were downloaded from the ESTIMATE tool (Estimation of Stromal and Immune cells in Malignant Tumor tissues using Expression data) [22]. The contribution of the mean expression of examined *TMEM* transcripts and the means of the immune, stromal, and ESTIMATE scores of HNSCC patients were compared using Chi-square test. These scores were used to define the infiltration of immune cells into tumor tissues as described previously by Yoshihara et al. Subpopulations of specific immune cells: lymphocytes, M1 and M2 macrophages, CD8 cells, naïve B cells, Th1 cells, Th2 cells, Th17 cells, and neutrophils were estimated using the supporting data presented by Thorsson et al. [23]. An influx of immune cell populations was compared with the low and high expression of analysed *TMEMs* using Mann–Whitney test. The analysis was performed as described in Section 2.5.

### 2.5. Statistical Analysis

All statistical analyses were performed using GraphPad Prism 8/9 (GraphPad, San Diego, CA, USA). The normality of data distribution was analyzed using the Shapiro–Wilk test. To compare expression levels of selected genes depending on clinical-pathological parameters, *t*-test or Mann–Whitney test were used. For expression analyses depending on tumor location, Kruskal–Wallis ANOVA with Dunn’s multiple comparison test, and ordinary one-way ANOVA with multiple comparison Tukey test were performed. Correlation between expression levels of *TMEMs* was measured using Pearson’s or Spearman’s correlation tests. The adequate statistical tests to compare groups were applied depending on the normality of data distribution. Survival rates were measured using the log-rank (Mantel–Cox) test and the Gehan-Breslow-Wilcoxon test with 95% confidence interval (CI) ratio defined using Mantel–Haenszel and log-rank tests. For all defined data and in all performed analyses, *p*-value < 0.05 was considered significant.

### 2.6. Validation of the Results

To validate the results obtained from the TCGA data analysis, the Gene Expression Omnibus (GEO) data repository, with GSE30784, gcRMA normalized [24] and GSE65858, log2-transformed and normalized using RSN was applied [25]. The IDs and symbols of genes were verified using the Genomic Scape portal (http://genomicscape.com/microarray/expression.php, accessed on 20 January 2021). The expression levels of the selected *TMEMs* were first compared between control (*n =* 16), dysplastic (*n =* 44), and cancer (*n =* 167) samples from oral localization. Next, *TMEM* expression levels were compared between HPV(−) and HPV(+) (*n =* 176 vs. 94) samples. The expression levels of *TMEMs* depending on the molecular cluster category including “atypical IR1” (*n =* 73) vs. “basal 4” (*n =* 84) vs. “classical 2” (*n =* 30) vs. “mesenchymal 3” (*n =* 83) were analyzed as described previously [25]. Finally, all patients (HPV(−) and HPV(+)) were divided into two subgroups using the mean of *TMEM* expressions as the cut-off and OS were calculated. GSE65858 set included HNSCC samples with cavum oris, hypopharynx, larynx, and oropharynx localizations. The statistical analysis was performed as described in Section 2.5.

## 3. Results

### 3.1. Expression of TMEMs Is Dysregulated in HNSCC Tumor Samples

The expression levels of 22 *TMEM*s in tumor vs. normal tissue were analyzed. Significant differences (*p* < 0.05) were observed for 16 analyzed genes. In the group of upregulated genes were *ANO1*, *TMEM17*, *TMEM48*, *TMEM97*, *TMEM140*, *TMEM156*, *TMEM158*, *TMEM206*, *RTP3*, *TMEM22*, and *TMEM43*. *TMEM45B*, *TMEM173*, *TMEM61*, *TMEM116*, and *TMEM213* were indicated as downregulated in HNSCC patient samples. For *TMEM25*, *TMEM30B*, *TMEM45A*, *TMEM88*, *TMEM98*, and *TMEM176A* no significant changes were observed (*p* > 0.05), Figure 1A.

The expressions of those 16 TMEM genes were then correlated in between each other. The gene with the highest number of significant correlations was *TMEM97*. It showed positive correlations with *TMEM17*, *TMEM22*, *TMEM48*, *TMEM116*, *TMEM158*, *TMEM206*, and *TMEM213*. Negative correlations were observed for *TMEM97*, *ANO1*, *TMEM45B*, *TMEM61*, *TMEM140*, and *TMEM173*, Figure 1B.

To define whether *TMEMs* have a potential to discriminate between healthy and tumor samples, the Receiver Operating Characteristic (ROC) curve analysis was performed. The largest areas under the curve (AUC) were shown for *TMEM206* (AUC = 0.9164, *p* < 0.0001), *TMEM97* (AUC = 0.8259, *p* < 0.0001), *TMEM158* (AUC = 0.8148, *p* < 0.0001), *ANO1* (AUC = 0.7439, *p* < 0.0001), *RTP3* (AUC = 0.7436, *p* < 0.0001), *TMEM45B* (AUC = 0.7969, *p* < 0.0001), *TMEM48* (AUC = 0.7180, *p* = 0.0005), *TMEM61* (AUC = 0.7996, *p* < 0.0001), and *TMEM213* (AUC = 0.6885, *p* = 0.0026) and the lowest for *TMEM140* (AUC = 0.6444, *p* = 0.0211), Figure 1C. There were no differences (*p* > 0.05) in the case of *TMEM17, TMEM22*, *TMEM43*, *TMEM116*, *TMEM156*, and *TMEM173*, Appendix A.

### 3.2. Expression of TMEMs Is Altered Depending on Clinicopathological Parameters

The expression levels of selected *TMEM*s were analyzed in the context of clinical-pathological parameters. The significant differences between the expression levels of *TMEM*s were indicated for A*NO1*, *TMEM156*, *TMEM158*, *TMEM206*, *TMEM173*, *TMEM116*, and *TMEM97* regarding four and more parameters, and for *TMEM45B*, *TMEM140*, *RTP3*, *TMEM22*, *TMEM43*, *TMEM61*, and *TMEM213* regarding less than four parameters; no statistical significances were observed for *TMEM17*, Figure 2. *TMEM43* expression correlated with age (*p* = 0.0064); *TMEM22*, *TMEM61*, *TMEM116*, and *TMEM97* expression correlated with gender (*p* = 0.0028, *p* = 0.0435, *p* = 0.0006, *p* = 0.0021, respectively), Figure 2.

The expression of *TMEM48*, *TMEM206,* and *TMEM22* correlated with HNSCC risk factors such as alcohol consumption (*p* = 0.0324, *p* = 0.0209, *p* = 0.0109, respectively); the expression of *ANO1, TMEM158, RTP3,* and *TMEM213* correlated with smoking (*p* = 0.0016, *p* = 0.0032, *p* = 0.0002, *p* = 0.0005, respectively).

We also assessed whether TMEMs are correlated with HPV status. The analysis showed significant correlations for *TMEM156*, *TMEM158*, *TMEM173*, and *TMEM116* (*p* < 0.0001) as well as for *ANO1, TMEM45B*, *TMEM206*, *RTP3*, and *TMEM97* (*p* = 0.002 for, *p* = 0.0347, *p* = 0.0198, *p* = 0.001, *p* = 0.0002, respectively).

Changes in the expression of *TMEMs* were frequently observed depending on T-stage and cancer grade. T3 and T4 tumors had higher levels of *TMEM156*, *TMEM158*, and *TMEM206* (*p* = 0.0336, *p* < 0.0001, *p* = 0.0364, respectively) and lower level of *TMEM97*, *TMEM140*, and *TMEM173* (*p* = 0.0153, *p* = 0.0185, *p* = 0.0035, respectively). For *TMEM22*, *TMEM97*, *TMEM116*, *TMEM140*, *TMEM156*, and *TMEM213* the elevated expression levels were associated with higher (G3 + G4) cancer grade (*p* = 0.0012, *p* = 0.0002, *p* = 0.0014, *p* = 0.0348, *p* < 0.0001, *p* = 0.0208, respectively) in contrast to *TMEM45B* and *TMEM158* which were highly expressed in G1 + G2 tumors (*p* < 0.0001 and *p* = 0.0367). 

*TMEM* expression was also dependent on tumor invasion to space surrounding a nerve for *ANO1*, *TMEM116*, and *TMEM97* (*p* = 0.002, *p* < 0.0001, *p* = 0.008) and on angiolymphatic invasion for *TMEM48*, *RTP3*, *TMEM173*, and *TMEM116* (*p* < 0.0001, *p* = 0.0267, *p* = 0.001, *p* = 0.0053, respectively). 

Regarding cancer stage and N-stage, there were significant correlations for *TMEM206*, *TMEM97* (*p* = 0.0479, *p* = 0.0075), and *TMEM45B* (*p* = 0.0011), respectively. 

We also analyzed the association of TMEMs with lymph nodes dissection status. We observed a positive correlation for *ANO1* and *TMEM158* (*p* = 0.0056 and *p* = 0.0008) and negative correlation for *TMEM48*, *TMEM116*, *TMEM156*, and *TMEM173* (*p* = 0.0096, *p* = 0.0063, *p* = 0.0049, *p* = 0.0001, respectively); detailed results are shown in Appendix A.

Next, the expression levels of selected *TMEM*s were compared for patients with different tumor localizations—within the oral cavity, pharynx, or larynx. Most of the analyzed genes showed statistically significant differences between the oral cavity and pharynx with the highest correlation (*p* < 0.0001) for *ANO1*, *TMEM22*, *TMEM97*, *TMEM116*, *TMEM156*, and *TMEM158*. *TMEM140*, *TMEM61*, and *TMEM43* showed no significant differences (*p* > 0.05). For other *TMEMs* statistically significant (*p* < 0.05) differences were indicated for at least two localizations, Figure 3.

### 3.3. Expression of ANO1, TMEM156, TMEM173, and TMEM213 Correlates with Patient Survival

We tested whether *TMEMs* are associated with patient survival. For each gene we measured disease-free survival (DFS) and overall survival (OS) in patients with high vs. low *TMEM* expression, Figure 4 and Appendix A.

In patients with lower expression of *ANO1* and *TMEM156* we observed significantly longer DFS (*p* = 0.0443, HR = 2.317 with 95% CI = 1.022 to 5.254 and HR = 2.415 with 95% CI = 1.066 to 5.470 and *p* = 0.0495, HR = 2.089 with 95% CI = 0.9207 to 4.741 and HR = 2.124 with 95% CI = 0.9363 to 4.817, respectively). Patients with low *ANO1* displayed also longer OS (*p* = 0.0003; HR = 1.710 with 95% CI = 1.280 to 2.284 and HR = 1.724 with 95% CI = 1.291 to 2.301). In contrast, lower levels of *TMEM156, TMEM173,* and *TMEM213* were associated with shorter OS (*p* = 0.0009, HR = 0.6123 with 95% CI = 0.4585 to 0.8177 and HR = 0.6043 with 95% CI = 0.4526 to 0.8068; *p* = 0.0494, HR = 0.7475 with 95% CI = 0.5592 to 0.9992 and HR = 0.7490 with 95% CI = 0.5605 to 1.001 and *p* = 0.0212, HR = 0.7853 with 95% CI = 0.5749 to 1.073 and HR = 0.7755 with 95% CI = 0.5679 to 1.059, respectively). Other genes showed no statistical differences in DFS or OS within the two groups of patients, Appendix A and Appendix A.

### 3.4. TMEM-Correlated Genes Are Involved in Numerous Biological Processes 

For further analysis we selected the four *TMEMs* which correlated with patient survival (*ANO1*, *TMEM156*, *TMEM173*, and *TMEM213)* and searched for the correlations with a panel of different genes, Figure 5A.

*ANO1* negatively correlated with genes related to metabolism, eukaryotic translation, and cellular response to stress and external stimuli; a positive correlation concerned the genes associated with homeostasis, extracellular matrix organization, and signaling by receptor tyrosine kinases. *TMEM156* was negatively correlated with genes implicated in processes such as *RHO GTPase* effectors and signaling and metabolism; it was positively correlated with the immune signaling and signaling by *GPCR*. *TMEM173* was negatively correlated with metabolism and processing of rRNA; *TMEM173*-positively correlated genes were related with the immune system and cytokine signaling. *TMEM213* negatively correlated with post-translational protein modification and cellular responses to external stimuli and positively correlated with the transport of small molecules and the immune signaling.

To assess the differences between patients with high and low expression of *TMEMs* at the functional level, the gene set enrichment analysis was performed. For *TMEM156* and *TMEM173* most gene sets were significantly (*p* < 0.05 and FDR < 0.25) enriched in high-expression phenotypes with regard to hallmarks (H) and oncogenic (C6) gene sets (Figure 5B).

High *ANO1* expression corresponded with the enrichment of the hallmarks signature genes (H) related to epithelial-mesenchymal transition, angiogenesis, and apical junction (NES = 1.896, 1.865, and 1.829, respectively). *TMEM156* and *TMEM173* displayed the high similarity in connection with apical surface and junction, apoptosis, and *IL2/STAT5*, *IL6/JAK/STAT5* (NES between 1.521 and 2.370). *ANO1*, *TMEM156*, and *TMEM173* were associated with the process of cell-to-cell apical junction formation. Genes associated with *MYC* targets and DNA repair were deregulated in the group of patients with lower *TMEM173* expression. A lack of significant enrichment of genes was indicated for patients both with low and high expression levels of *TMEM213*.

Next, the second set of genes that were grouped in the oncogenic signature (C6) were analyzed. For patients with higher expression of *ANO1*, *TMEM156,* and *TMEM173* the changes in the groups of genes (with NES between 1.642 and 2.469) which were similar to these induced by modification in the expression levels of important proto-oncogenes and suppressors such as *MYC*, *KRAS*, *CyclinD1*, and *P53* or genes associated with EMT process or maintaining of the cancer-initiating cells (CICs) population were indicated. Moreover, changes in patients with low expression of *TMEM173* were also observed and connected with phenotypes similar to those with upregulated *EIF4E* or *MYC* (NES = −1.744 and NES = −1.856, respectively). It was also indicated that both *TMEM156* and *TMEM173* had changed phenotypes similar to cells after knockdown of ribosomal protein S14 (*RPS14*) and regulator of chromatin (*SNF5)* genes (NES = 2.284 and NES = 2.195). In the case of the oncogenic signature, significant changes of genes were indicated for *TMEM213*. The reduction of genes (NES between −1.756 and −1.592) which were characteristic for the cells with modified expression of *CTNNB1*, *ERBB2*, *KRAS*, *LET2*, *NFE2L2,* or *STK33* as well as genes upregulated during the late stages of differentiation of embryoid bodies, stimulated with *IL21* or after *mTOR* pathway inhibition was observed for patients with higher expression of *TMEM213*. All data are presented in Figure 5B. The list of genes involved in the above pathways are presented in Appendix A and Appendix A.

### 3.5. Expression Levels of ANO1, TMEM156, TMEM173, and TMEM213 Are Associated with Immunological Indicators 

The GSEA analysis of samples with low vs. high expression of *ANO1*, *TMEM156*, *TMEM173*, and *TMEM213* showed the enrichment of immune (C7) gene sets. The next step was to perform patient immune profiling depending on *TMEM* expression levels, Figure 6. 

Patients with higher *TMEM173* expression displayed the enrichment of genes associated with CD8 and CD4 T cell activation (NES between 2.386 and 2.264) in contrast to the group with low *TMEM173* expression, where downregulation of genes related with macrophages, dendritic cells (DCs) and B cells, and the response of activated CD4 and CD8 T cells (NES between −1.815 and −2.058) was observed. Patients with high expression of *TMEM156* displayed gene profiles which were characteristic for changes between mast cells, CD4, and B cells in comparison to NK cells as well as profiles associated with the specified response of monocytes, CD4, macrophages, and neutrophils in response to different stimuli (NES between 2.514 and 2.585). In patients with low expression of *ANO1* changes in profiles describing differentiated naive T cells and DCs populations were indicated (NES = −2.06). The results are presented in Figure 6A. 

Next, HNSCC patients with low or high levels of specific *TMEMs* were analyzed depending on the purity of samples using the immune, stromal, and ESTIMATE scores obtained from an online tool. The chi-squared tests showed that nine of the examined *TMEM* transcripts, including *ANO1*, *TMEM156*, and *TMEM173* were significantly changed regarding the immune score (*p* = 0.0089, *p* < 0.0001, and *p* < 0.0001, respectively). All data from this analysis are presented in Figure 6B and Appendix A.

The fractions of selected immune cells in high and low *TMEM* expression groups were analyzed. There was a significantly higher fraction of lymphocytes (general population), CD8 cells and naive B cells in a group with low expression of *ANO1* (*p* < 0.0001, *p* < 0.0001, and *p* = 0.0002, respectively). Patients with high expression of *TMEM156* and *TMEM213* displayed higher infiltration of naive B cells (*p* < 0.0001 and *p* < 0.0001, respectively) and these two *TMEMs* along with *TMEM173* high expression group showed increased infiltration of the general population of lymphocytes (*p* < 0.0001, *p* = 0.0003, and *p* < 0.0001, respectively). High expression of *TMEM156* and *TMEM173* also correlated with elevated fractions of CD8 and Th1 cells (*p* < 0.0001 and *p* < 0.0001 as well as *p* < 0.0001, and *p* < 0.0001, respectively). Higher fractions of M1 macrophages were observed in patients with high expression of *TMEM173* (*p* = 0.0203) and low expression of *TMEM213* (*p* = 0.0259). Macrophages with M2 phenotype were enriched in patients with high expression of *ANO1* (*p* = 0.0153) and low expression of *TMEM173* (*p* = 0.2870). Increased infiltration of Th17 cells was indicated in patients with high expression of *TMEM173* (*p* = 0.0066), Figure 6C.

### 3.6. Validation of Results for Selected TMEMs Using the GEO Data

Validation of the results for *ANO1, TMEM156, TMEM173,* and *TMEM213* was made based on the GSE30784 and GSE65858 datasets. There was significant upregulation of *ANO1* expression in tumor vs. dysplastic and healthy tissues (*p* < 0.0001 and *p* = 0.0315, respectively). In addition, upregulation of *TMEM173* in tumor vs. healthy tissue was observed (*p* = 0.0002). No differences for *TMEM156* and *TMEM213* were indicated (*p* = 0.9476 and *p* = 0.1539, respectively), Figure 7A. 

Next, the expression levels of *TMEMs* depending on HPV status were tested, and downregulation of *ANO1* and upregulation of *TMEM156* and *TMEM173* were indicated in HPV(+) compared to HPV(−) patients (*p* < 0.0001, *p* = 0.0040 and *p* = 0.0017, respectively). No correlation between the expression of *TMEM213* and HPV status was noticed (*p* = 0.5036), Figure 7B. Based on the molecular classification presented by Wichmann et al., it was indicated that expression levels of *ANO1* were the highest in the “classical 2” and the “mesenchymal 3” subtypes (*p* < 0.0001), in contrast to *TMEM156* which displayed the highest expression in the “atypical IR1” compared to the rest of the subtypes (*p* < 0.0001). *TMEM173* displayed the highest expression in the “atypical IR1” and the lowest in the “classical 2” subtype (*p* < 0.0001). No differences between four molecular subtypes of HNSCC and expression levels of *TMEM213* were noticed (*p* = 0.9527) (Figure 7B). In the last validation analysis, the overall survival was assessed. No differences between patients’ survival and expression levels of *ANO1*, *TMEM173*, or *TMEM213* were observed in both short (3 years) (*p* = 0.2795, *p* = 0.2431 and *p* = 0.4280, respectively) and long-time (up to 7 years) monitoring (*p* = 0.4408, *p* = 0.3380 and *p* = 0.7975, respectively). However, patients with higher levels of *TMEM156* displayed significantly longer OS than patients with lower *TMEM156* expression in both short-time and long-time monitoring (*p* = 0.0189 and *p* = 0.0003, respectively), Figure 7C.

Next, the overall survival was correlated with the HPV status. No significant (*p* > 0.05) association between OS and *ANO1*, *TMEM156*, and *TMEM213* expression in HPV(−) patients was observed. However, HPV(−) patients with higher levels of *TMEM173* displayed better OS than patients with lower levels of this gene (*p* = 0.0406, HR = 0.6808, 95% CI = 0.4712 to 0.9836 and HR = 0.7000, 95% CI = 0.4890 to 1.002). In HPV(+) patients a strong association between higher expression of *TMEM156* and longer OS time was observed (*p* < 0.0001, HR = 0.2778, 95% CI = 0.1485 to 0.5198 and HR = 0.3709, 95% CI = 0.2086 to 0.6594). For the rest of the analyzed *TMEMs,* no significant (*p* > 0.05) differences in the group of HPV(+) patients were observed, Figure 7C.

## 4. Discussion

Transmembrane (TMEM) proteins are related with multiple steps of development and progression of several cancer types [5] such as head and neck squamous cell carcinoma [26,27], lung cancer [28,29,30] colorectal cancer [31,32], breast cancer [33,34,35], and renal cancer [36,37]. Despite their importance in carcinogenesis, their biological role is still poorly characterized, thus their diagnostic utility is still limited. To our knowledge, this is the first comprehensive analysis of the selected cancer-related TMEMs [5,6,9] in patients with HNSCC. 

In this study we performed a panel of in silico analyses of TCGA data to characterize the role of selected TMEMs in head and neck squamous cell carcinoma (HNSCC). Among twenty-two analyzed TMEM genes four of them: *ANO1*, *TMEM156*, *TMEM173,* and *TMEM213*, showed altered expression levels in the neoplastic compared to normal tissue and statistically significant differences in the survival rates. Our main findings are as follows:*ANO1* and *TMEM156* are upregulated in tumor tissue, while *TMEM213* and *TMEM173* are downregulated in HNSCC. Patients with low *ANO1* and *TMEM156* expression display higher probability to stay free from disease recurrence. Moreover, longer overall survival can be expected for patients with high *TMEM156*, *TMEM173*, *TMEM213* expression and with low *ANO1* expression.Patients with high *TMEM140*, *TMEM156*, and *TMEM173* expression levels have different transcription profiles compared to other patients.The genes correlated with *TMEMs* are involved in numerous biological processes, including those associated with immune response. The expression of *TMEM17*, *TMEM97*, *TMEM140*, *TMEM156*, *TMEM173*, and *TMEM206* show significant correlation with immune, stromal, and ESTIMATE scores which indicates a strong association between those *TMEMs* and immune response inside a tumor microenvironment.

Noteworthy, only *ANO1* [26,38,39,40] and *TMEM173* [27] have been previously described in HNSCC. We noticed that expression levels of *TMEM156* and *TMEM213* were negatively correlated with *ANO1*, while *TMEM173* and *TMEM213* positively correlated with *TMEM156*. This observation supports the hypothesis that *ANO1* may play the opposite role compared to other described *TMEMs* in HNSCC patients. Previously the associations between *TMEM213* and *TMEM30B* and between *TMEM72* and *TMEM116* have been described, however, for clear cell renal cell carcinoma [36].

According to ROC analysis and due to the significant correlation of *TMEM* expression with patient survival (OS and DFS), our study showed that these *THEMs* could serve as prognostic and diagnostic biomarkers in HNSCC. In meta-analysis involving 1760 patients from 7 independent studies, Zhang et al. demonstrated the potential prognostic value of *ANO1* in multiple cancer types, including HNSCC [41]. Although higher *ANO1* expression is associated with worse OS and DFS, an *in vitro* model of HNSCC demonstrated that overexpression of *ANO1* in Te11 cells was linked with enhanced sensitivity to *EGFR*-targeted agent, Gefitinib, making *ANO1* a good predictive biomarker and potential target for *EGFR*-directed therapy [42]. It has also proven its diagnostic utility in other cancer types [43,44,45,46]. In line with our findings, expression of *TMEM156* had an impact on patient survival, but it did not have significant potential to differentiate neoplastic tissue from the healthy one. To our knowledge, the diagnostic usefulness of *TMEM156* has not been proven in any previous study. *TMEM173* has been previously described as a prognostic biomarker only in hepatocellular carcinoma, where its decreased expression was associated with worse outcome [47,48]. According to our analysis, *TMEM173* can serve as a biomarker of prognosis also in patients with HNSCC, where higher expression significantly correlates with longer OS. Little is known about the diagnostic role of *TMEM213*. Zou et al. proved that *TMEM213* can act as an independent prognostic and predictive marker in non-small cell lung cancer patients (NSCLC) after surgical resection [30]. Our study indicated that *TMEM213* could serve as a prognostic biomarker of survival also in HNSCC since patients with its higher expression showed worse OS. Both *TMEM173* and *TMEM213* did not show the ability to distinguish between healthy and neoplastic tissue, thus they could not serve as diagnostic biomarkers in the case of HNSCC.

To validate the results for *ANO1*, *TMEM156*, *TMEM173,* and *TMEM213,* we used the two GEO datasets. Taking into account comparison between healthy and cancer samples only in the case of *ANO1* and *TMEM173*, we observed the same significant changes as in our data obtained from the TCGA. No differences were noticed for *TMEM156* or *TMEM213*. However, it should be emphasized that the GSE30784 dataset [24] used in this work represents only samples from oral localization in contrast to the TCGA data where we included samples of oral cavity, pharynx, and larynx localizations. Next, based on GSE65858 datasets [25], we validated changes in *ANO1*, *TMEM156*, *TMEM173,* and *TMEM213* depending on HPV status and patient survival. Little is known about HPV infection and changes in *TMEM* expression, thus the validation of our results based on different patient data would be of particular importance.

For *TMEM156*, *TMEM173*, and *TMEM213* we noticed (based on both datasets) upregulation of *TMEM156*, *TMEM173* and no changes for *TMEM213* between HPV(+) and HPV(−) patients. Surprisingly, in the TCGA analysis, *ANO1* was upregulated, in contrast to the GEO results, where we observed its downregulation in HPV(+) samples. 

*ANO1* is the best described *TMEM* in HNSCC. Ayoub et al. indicated that *ANO1*, like other genes on the 11q13, was amplified and overexpressed in HNSCC patients. Most likely *ANO1* is responsible for distant metastasis formation by regulation of cell migration. Blocking the calcium-activated chloride channel activity of *ANO1* leads to reduction of cell migration, which makes it a new potential target of therapy [49]. Dixit et al., indicated overexpression of *ANO1* in HPV(−) HNSCC samples based on immunohistochemistry staining of clinical samples and TCGA data [50]. *ANO1* was identified as one of the downregulated genes in HPV(+) HNSCC patients [51]. It should be noted that the 11q13 region containing *ANO1* is characteristic of HPV-negative cancers [52,53].

In the case of *TMEM173,* Liang et al. observed no differences between HPV(+) and HPV(−) patients based on IHC staining of 50 patient samples. However, they indicated that *TMEM173* was presented as an activated form much more frequently in the HPV(+) group than in HPV(−) group [27]. In our study, the OS was analyzed for all HNSCC patients and only for *TMEM156* significant differences were noticed in both GEO and TCGA analyses. It must be noted that GSE65858 datasets represent only samples from cavum oris, hypopharynx, larynx, and oropharynx localizations, and the number of samples is nearly half smaller and collected from one population [25] in contrast to the TCGA cohort. However, when the patients from GEO were divided depending on HPV status, the association between longer OS time and *TMEM156* expression was indicated for only HPV(+) patients (and not for HPV(−)), which proves the connection of this *TMEM* with a viral infection. Moreover, for high *TMEM173* levels only the group of HPV(−) patients showed better survival, which is in line with our results based on all sets of HNSCC patients from the TCGA data, where only 5.5% of all analyzed patients were HPV positive in this group.

To address the issue of the biological role of *TMEMs* in carcinogenesis, we performed the analysis of the correlated gene sets. Our results showed that genes positively correlated with *ANO1* were involved in a cellular matrix organization as well as receptor tyrosine kinase signaling. This observation supports the fact that overexpression of *ANO1* contributes to metastasis by promoting the epithelial-mesenchymal transition (EMT) process by deregulation of several signaling pathways like *MAPK, EGFR,* or *CAMK* [28,54]. On the other hand, genes negatively correlated with *ANO1* were involved in cellular response to stress and external stimuli, which may favor further DNA damage and thus contribute to cancer progression. However, the exact role of *ANO1* has not been previously described, so it requires further examination. It is only known that *ANO1* was linked with cancer metastasis, but the molecular basis of this process remains unknown [49]. In addition, the role of *TMEM156* in cancer is poorly understood. Cheishvili et al. listed *TMEM156* as one of the invasion-promoting genes in an *in vitro* model of prostate, breast, and liver cancers [55]. Contrarily, in line with our findings, patients with higher *TMEM156* may have better immune responses since many genes positively correlated with *TMEM156* are involved in multiple immunological processes. What is more, genes negatively correlated with *TMEM156* are linked with *RHO GTPase* effectors, which results in a poorer response to receptor activation, thus reducing cancer cell proliferation, survival, and migration [56]. *TMEM173* is a well-known regulator of the innate immune response [57] and initiator of an immune response against tumors in several cancer types [58] which align with our results. Genes positively correlated with *TMEM173* are involved not only in promoting innate immunity response but also mechanisms of adaptive immunity. Despite the fact that *TMEM213* expression is altered in several cancer types, its role in cancer development and progression, to our knowledge, still remains unknown. Bioinformatic analysis performed by Zou et al. revealed that high *TMEM213* expression was associated with resistance to paclitaxel in patients with lung adenocarcinoma, which was related to cytochrome P450 and ABC transporters [30]. Our results showed that genes positively correlated with *TMEM213* are also involved in the innate immune response. That may suggest the important role of *TMEM213* in the immune response against the tumor; however, these results are not as conclusive as in the case of *TMEM156* and *TMEM173*. 

The GSEA analysis was performed to further investigate the biological role of *TMEMs* depending on their expression levels. The hallmarks analysis for *ANO1* confirmed its participation in the EMT process by deregulation of genes directly involved in EMT as well as in apical junction formation. It is worth mentioning that the presence of *ANO1* was found in the apical membrane of epithelial cells in the airways and gastrointestinal tract [39]. *ANO1* has a proven role in metastasis via association with Hedgehog signaling [59], which is consistent with our GSEA results. Overexpression of *ANO1* may also promote metastasis through the involvement in angiogenesis, which is also confirmed by the fact that genes positively regulated with *ANO1* are involved in collagen formation. That has become another previously unknown mechanism by which *ANO1* may influence tumor progression. Analysis for *TMEM156* also confirmed the findings from the REACTOME analysis. What is more, the gene sets enriched in HNSCC patients with high *TMEM156* expression are similar to those with high expression of *TMEM173*, which may suggest a similar function in cancer development and progression. Myc targets are the gene sets enriched in patients with low expression level of *TMEM173*, which may explain the worse OS of those patients [60]. Moreover, in the validation of *ANO1*, *TMEM156*, *TMEM173* based on GEO datasets from the study presented by Wichmann et al. [25], we indicated that the expression level of *ANO1* is the lowest in “atypical IR 1” and the highest in “mesenchymal 3” cluster of HNSCC. In contrast, we observed the reversed expression levels of *TMEM156* and TMEM173 in molecular subtypes of HNSCC. Wichmann et al. showed that “atypical IR 1” has characteristics mainly for oropharyngeal cancers, wild type of *TP53*, and with HPV infection. Moreover, this molecular cluster was indicated as enriched with genes connected with the cell cycle and immune response. It should be noted that the “mesenchymal 3” cluster was also connected with the worst prognosis [25]. These results underline the oncogenic function of *ANO1* and suppressor roles of *TMEM156* and *TMEM173.*

Due to the fact that described *TMEMs* seem to be related to immunological processes, the infiltration levels of immune cells in tumor tissue was evaluated. Such analysis has never been performed for *ANO1*, *TMEM156*, and *TMEM213* in HNSCC and may help to further understand the influence of described *TMEMs* on the immunological processes during carcinogenesis. It has been proven that activation of *TMEM173*, its translocation from the membranes of the endoplasmic reticulum and mitochondria to the cytoplasm, enhances regulatory T cells infiltration in HPV(−) positive HNSCC patients with tumors localized in the oral cavity [27]. However, the effect of high expression of *TMEM173* has not been demonstrated. 

HNSCC patients with higher *ANO1* expression have fewer immune cells in the tumor microenvironment, which may be explained by the involvement of cytokine signaling-related genes, which were positively regulated with this *TMEM*. Contrarily, immune cells infiltrate tumors more in patients with higher *TMEM156* and *TMEM173* levels. *TMEM213* showed to be not involved in the modeling of tumor microenvironment.

Depending on a subtype, lymphocytes may participate both in cancer promoting and antitumor response [61]. Oppositely to *TMEM156*, *TMEM173*, and *TMEM213*, patients with higher *ANO1* expression have a lower fraction of lymphocytes in the tumor microenvironment. This may explain the worst impact of *ANO1* overexpression on patient survival. We also analyzed the differences in macrophage phenotypes depending on the TMEMs level. Phenotypes of M1 and M2 macrophages are plastic and they should be defined by their gene expression profiles [62] like it was made by Thorsson et al. [23] whose results were used in this study. M1 macrophages, activated by *IFNγ*, are capable of priming antitumor immune response by proinflammatory cytokines such as *TNFα*, *IL-1*, or *IL-6* release [62]. We observed the differences in the amount of M1 macrophages only for *TMEM173* and *TMEM213*. A higher amount of M1 macrophages was observed in patients with higher *TMEM173* expression. It is worth mentioning that based on GSEA those patients have enriched genes related to *IFNγ* response. On the other hand, M2 or “alternatively” activated macrophages are considered as tumor-associated macrophages (TAMs), which mostly promote tumor growth and may be obligatory for cancer progression (angiogenesis, invasion, and metastasis) and their high content generally correlates with poor prognosis [62]. Our results indicated that patients with higher expression of *ANO1* and lower expression of *TMEM173* exhibited a higher influx of M2 macrophages. It should be noted that these patients also displayed significantly shorter OS. Next, CD8 cells which exhibit antitumor properties [62] were analyzed depending on *TMEM* levels. HNSCC patients with higher *ANO1* expression had a lower fraction of CD8 cells and this fraction was higher in patients with higher *TMEM156* and *TMEM173*. It has been previously indicated that CD8 cells infiltrating tumors possess prognostic value, and patients with higher levels of those cells benefited from neoadjuvant chemoradiation in the case of rectal cancer [63]. According to our results, patients with lower *ANO1* expression as well as higher *TMEM156* and *TMEM213* expression had a smaller fraction of naïve B cells. The opposite occurrence of this cell type is caused by their dual role (both anti- and pro-tumorigenic) in cancer. They can facilitate response against tumor cells upon activation and conversion to antigen-presenting B cells. Naïve resting B cells may initiate the state of unresponsiveness in naïve CD4+ or CD8+ T cells [64]. It has been also noted that cancer cells may attract naïve B cells and promote their differentiation into Breg cells, which are able to facilitate cancer progression [65]. Finally, the disproportion in some Th cell types was observed between patients with low and high expression of *TMEM156*, *TMEM173,* and *TMEM213*. Th1 cells, which stimulate cytotoxic T lymphocytes (CTLs) to facilitate tumor rejection [62], are present to a greater extent in patients with high expression of *TMEM156* and *TMEM173*. This also supports the hypothesis that those patients’ immune system is better adapted to fight cancer and thus higher expression of those genes correlates with longer survival. A similar situation occurs also with regard to Th17, which exhibits antitumor properties by activating CTLs [62], in patients with higher expression of *TMEM156*. 

It should be emphasized that our results are based on the whole set of HNSCC patients without excluding HPV positive cases and presented the general landscape of *TMEMs* in this type of cancer. We decided to analyze the whole set of patients because the TCGA data did not contain full information about HPV status; only 39/522 cases were described as HPV(+) and 79/522 as HPV(−) samples. However, those data were sufficient to show that the expression of certain *TMEMs* depended on HPV infection. Further examination of *TMEMs* in the context of viral infection will shed new light on the biological role of these genes and their potential influence on targeted therapies, chemoradiotherapy [66], or even immunotherapy [67].

## 5. Conclusions

This study gives a detailed insight into *TMEMs*’ role in cancer development and progression. It demonstrated a significant potential of *ANO1, TMEM156, TMEM173, TMEM213* as biomarkers in HNSCC. Low expression of *ANO1* and *TMEM156* correlated with longer disease-free survival and high *TMEM156, TMEM173, TMEM213*, as well as low *ANO1* expression correlated with longer overall survival. We proposed a new role of *ANO1* as the oncogene in HNSCC pathogenesis. Moreover, we showed that the *TMEM* genes are involved in numerous cellular processes and emphasized the positive correlation of *TMEMs* with the immune response. Since our work was based on in silico analyses of TCGA data, further studies on in vitro, in vivo, or clinical models should confirm the results presented here.

## Figures and Tables

**Figure 1 cancers-13-04737-f001:**
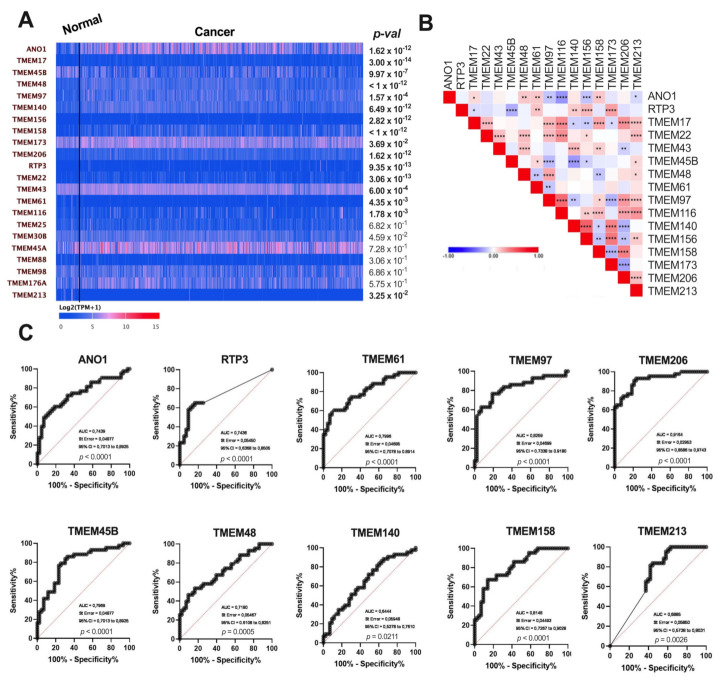
(**A**) Expression levels of selected *TMEM*s genes in tumor and normal tissue in patients with HNSCC; data from UALCAN presented as log2(TPM + 1); modified; (**B**) Heatmap illustrating correlation between expression levels of 16 selected *TMEM*s in HNSCC patients; the color of each square indicates the R coefficient value; Pearson’s and Spearman’s correlation tests; * *p* < 0.05; ** *p* < 0.01; *** *p* < 0.001; **** *p* < 0.0001; empty squares indicate no significant correlation; (**C**) Receiver operating characteristic curve (ROC) analysis of statistically significant *ANO1*, *RTP3*, *TMEM45B*, *TMEM48*, *TMEM61*, *TMEM97*, *TMEM140*, *TMEM158*, *TMEM206,* and *TMEM213* of HNSCC samples and paired adjacent normal tissues.

**Figure 2 cancers-13-04737-f002:**
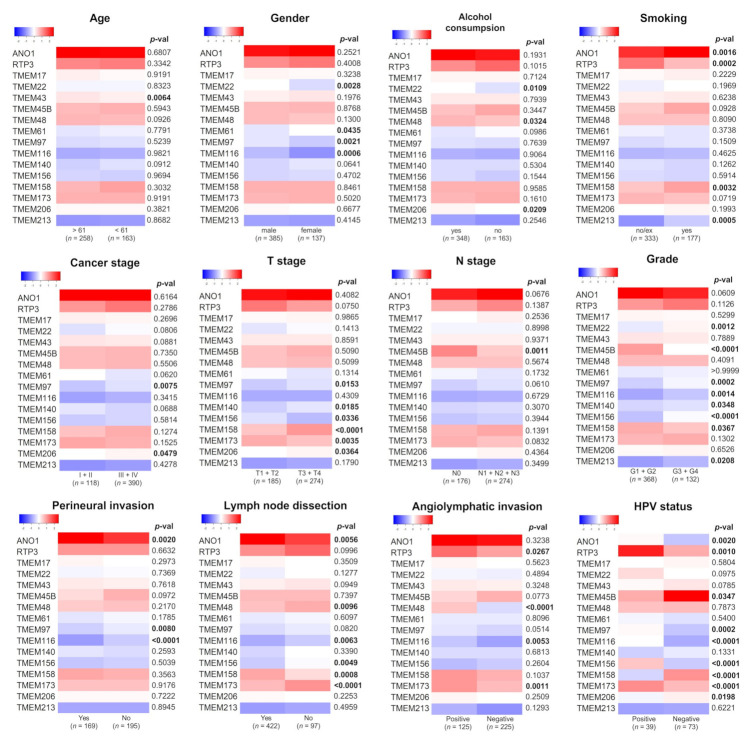
Comparison of selected *TMEM*s expression depending on clinicopathological parameters in patients with HNSCC. Significantly changed *p*-values (*p* < 0.05) were bolded; *n*—number of cases in each subgroup.

**Figure 3 cancers-13-04737-f003:**
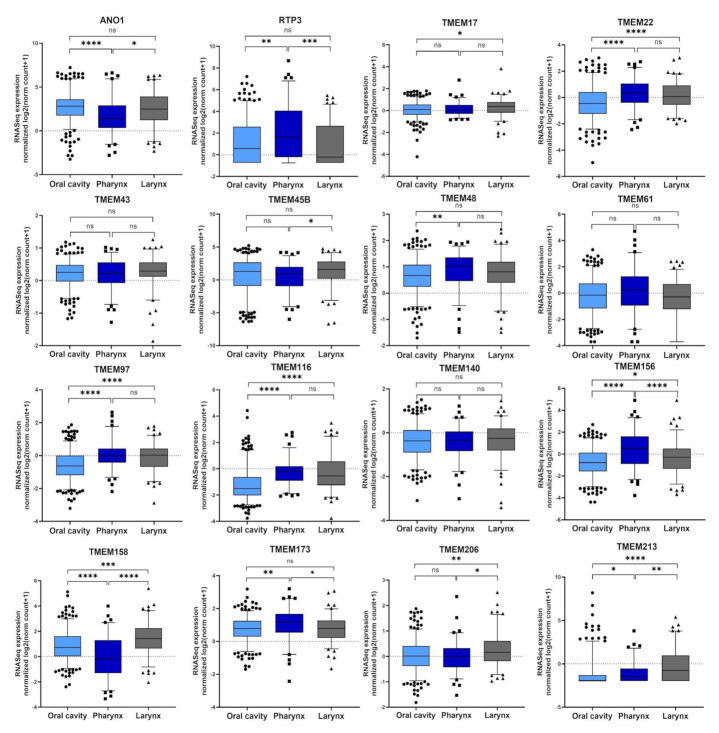
The expression levels of selected *TMEMs* genes depend on head and neck squamous cell carcinoma (HNSCC) localization in oral cavity (*n =* 316), pharynx (*n =* 90), and larynx (*n =* 116); *ns*: not significant; * *p* < 0.05; ** *p* < 0.01; *** *p* < 0.001; **** *p* < 0.0001.

**Figure 4 cancers-13-04737-f004:**
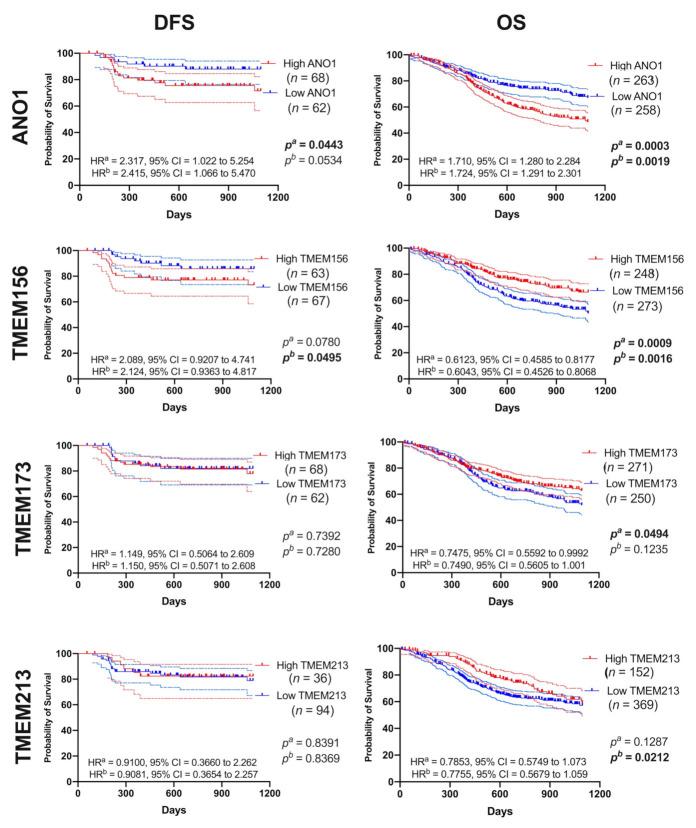
Disease-free survival (DFS) and overall survival (OS) curves (darker lines) with 95% CI (lighter lines) of HNSCC patients depending on *ANO1* or *TMEM156* or *TMEM173* or *TMEM213* expression levels; high and low subgroups of patients divided based on the mean of expression level; *n*—number of cases in each subgroup; *p^a^*—log-rank (Mantel–Cox) test, and *p^b^*—Gehan-Breslow-Wilcoxon test; HR^a^—Hazard Ratio-Mantel–Haenszel, HR^b^—Hazard Ratio-log-rank; CI —confidence interval; *p* < 0.05 considered as significant.

**Figure 5 cancers-13-04737-f005:**
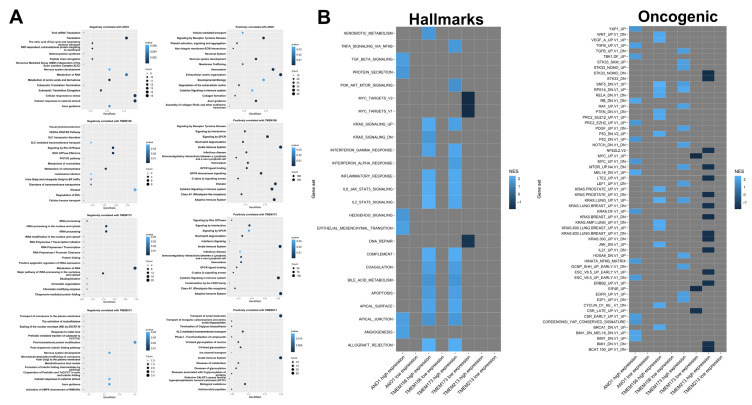
Differences in HNSCC phenotype depending on the expression level of *ANO1*, *TMEM156*, *TMEM173,* and *TMEM213*. (**A**) REACTOME pathway enrichment analysis of genes positively (R > 0.3) and negatively (R < −0.3) correlated with *ANO1*, *TMEM156*, *TMEM173,* and *TMEM213*. Fifteen statistically significant (*p* < 0.05) pathways with the highest gene count are shown; (**B**) Genes sets significantly (*p* < 0.05 and FDR < 0.25) enriched in high and low *ANO1*, *TMEM156*, *TMEM17,* and *TMEM213* expression phenotypes with regard to hallmarks (H) and oncogenic (C6) gene sets based on GSEA analysis; gray squares indicate lack of enrichment.

**Figure 6 cancers-13-04737-f006:**
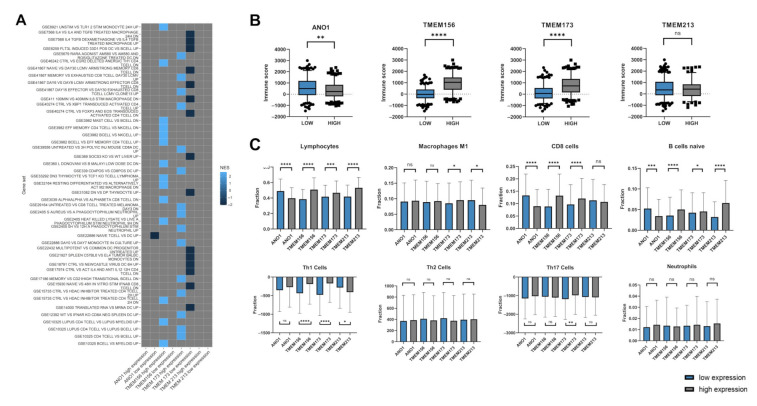
Immunologic profile of HNSCC patients depending on expression levels of *ANO1*, *TMEM156*, *TMEM173,* and *TMEM213*; (**A**) phenotype of patients depending on TMEMs levels based on GSEA analysis regarding to immunologic (C7) gene sets with *p* < 0.05 and FDR < 0.25; gray squares indicate lack of enrichment; (**B**) the level of immune score for low and high expression level of selected *TMEM*s with the significant p-value. *ns*: not significant. (**C**) Comparison between low and high expression of *TMEMs* for lymphocytes, CD8 cells, macrophages M1 and M2, B cells naive, Th1 cells, Th2 cells, Th17 cells and neutrophils with the significant *p*-value; * *p* < 0.05; ** *p* < 0.01; *** *p* < 0.001; **** *p* < 0.0001.

**Figure 7 cancers-13-04737-f007:**
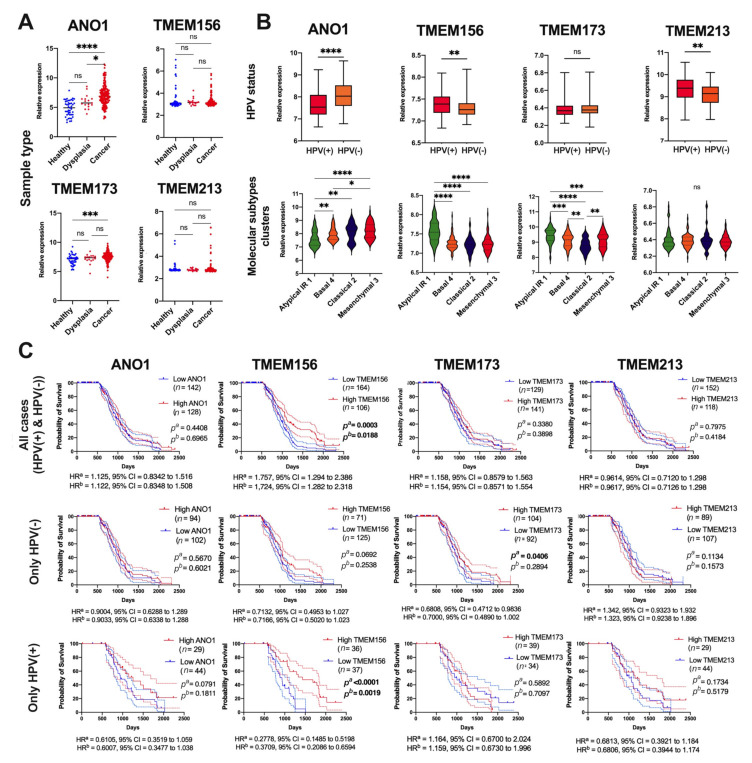
Validation of *ANO1*, *TMEM156*, *TMEM173,* and *TMEM213* in HNSCC patients using GEO datasets: (**A**) expression level of *TMEM*s depending on sample types, based on GSE30784; (**B**) expression level of *TMEM*s depending HPV status, and *TMEM*s levels in different types of HNSCC divided into molecular clusters and (**C**) OS of HNSCC patients depending on *TMEMs* levels in all cases, only in HPV(−) and only in HPV(+) patients, based on GSE65858; Mann–Whitney U or *t*-test or one-way ANOVA test with post-test; *ns*: not significant, * *p* < 0.05, ** *p* < 0.01, *** *p* < 0.001, **** *p* < 0.0001; *p^a^*—log-rank (Mantel–Cox) test, *p^b^*—Gehan-Breslow-Wilcoxon Test; HR^a^—Hazard Ratio-Mantel–Haenszel, HR^b^—Hazard Ratio-logrank; CI—confidence interval; *p* < 0.05 considered as significant.

## Data Availability

The datasets used and/or analyzed during the current study are available from the corresponding author on reasonable request. Raw data are available on the XenaBrowser, Ualcan, cBioportal, and ESTIMATE databases.

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
