# Peer review of "The Landscape of Transmembrane Protein Family Members in Head and Neck Cancers: Their Biological Role and Diagnostic Utility"

_cancers, 2021, doi:10.3390/cancers13194737_

Round 1

Reviewer 1 Report

The authors carefully revised their manuscript.

However, I think Fig. 5, 6 and 7C are still room for improvement.

Author Response

Dear Editor and Reviewers,

Thank you very much for your revisions and suggestions made for manuscript: The Landscape of Transmembrane Protein Family Members in Head and Neck Cancers: Their Biological Role and Diagnostic Utility, ID: cancers-1366944 . We have read them in detail and most of them were used by us to improve our manuscript. All of the changes were included in the current version of the draft and marked. The detailed responses point by point to the suggestions are included below in and in the text.

We hope that the corrections made are in line with your suggestions and expectations, and are enough to publish our publication in the Cancers MDPI journal. 

We are glad that we could cooperate with you under the presented topic: diagnostic utility and biological role of TMEMs in HNSCC.

Best regards

Authors of manuscript

Review Report Form 

Open Review

(x) I would not like to sign my review report  

( ) I would like to sign my review report  

English language and style

( ) Extensive editing of English language and style required  

( ) Moderate English changes required  

( ) English language and style are fine/minor spell check required  

(x) I don't feel qualified to judge about the English language and style  

Yes

Can be improved

Must be improved

Not applicable

Does the introduction provide sufficient background and include all relevant references?

(x)

( )

( )

( )

Is the research design appropriate?

(x)

( )

( )

( )

Are the methods adequately described?

(x)

( )

( )

( )

Are the results clearly presented?

( )

(x)

( )

( )

Are the conclusions supported by the results?

(x)

( )

( )

( )

Comments and Suggestions for Authors

The authors carefully revised their manuscript.

However, I think Fig. 5, 6 and 7C are still room for improvement.

Responses: The English spelling and grammar was improved and changed in the revised manuscript.

The figures were improved and submitted in better quality - 600 dpi. All figures are in the line of MDPIs' requirements.

Review Report Form 

Open Review

(x) I would not like to sign my review report  

( ) I would like to sign my review report  

English language and style

( ) Extensive editing of English language and style required  

(x) Moderate English changes required  

( ) English language and style are fine/minor spell check required  

( ) I don't feel qualified to judge about the English language and style  

Yes

Can be improved

Must be improved

Not applicable

Does the introduction provide sufficient background and include all relevant references?

(x)

( )

( )

( )

Is the research design appropriate?

(x)

( )

( )

( )

Are the methods adequately described?

(x)

( )

( )

( )

Are the results clearly presented?

(x)

( )

( )

( )

Are the conclusions supported by the results?

(x)

( )

( )

( )

Comments and Suggestions for Authors

Thank you for your attention to detail with the revised manuscript. The figures are more clear with the exception of Figure 5 and 7C which are difficult to read. 

Responses: The English spelling and grammar was improved and changed in the revised manuscript.

The figures were improved and submitted in better quality - 600 dpi. All figures are in the line of MDPIs' requirements.

Reviewer 2 Report

Thank you for your attention to detail with the revised manuscript. The figures are more clear with the exception of Figure 5 and 7C which are difficult to read. 

Author Response

Dear Editor and Reviewers,

Thank you very much for your revisions and suggestions made for manuscript: The Landscape of Transmembrane Protein Family Members in Head and Neck Cancers: Their Biological Role and Diagnostic Utility, ID: cancers-1366944 . We have read them in detail and most of them were used by us to improve our manuscript. All of the changes were included in the current version of the draft and marked. The detailed responses point by point to the suggestions are included below in and in the text.

We hope that the corrections made are in line with your suggestions and expectations, and are enough to publish our publication in the Cancers MDPI journal. 

We are glad that we could cooperate with you under the presented topic: diagnostic utility and biological role of TMEMs in HNSCC.

Best regards

Authors of manuscript

Review Report Form 

Open Review

(x) I would not like to sign my review report  

( ) I would like to sign my review report  

English language and style

( ) Extensive editing of English language and style required  

( ) Moderate English changes required  

( ) English language and style are fine/minor spell check required  

(x) I don't feel qualified to judge about the English language and style  

Yes

Can be improved

Must be improved

Not applicable

Does the introduction provide sufficient background and include all relevant references?

(x)

( )

( )

( )

Is the research design appropriate?

(x)

( )

( )

( )

Are the methods adequately described?

(x)

( )

( )

( )

Are the results clearly presented?

( )

(x)

( )

( )

Are the conclusions supported by the results?

(x)

( )

( )

( )

Comments and Suggestions for Authors

The authors carefully revised their manuscript.

However, I think Fig. 5, 6 and 7C are still room for improvement.

Responses: The English spelling and grammar was improved and changed in the revised manuscript.

The figures were improved and submitted in better quality - 600 dpi. All figures are in the line of MDPIs' requirements.

Review Report Form 

Open Review

(x) I would not like to sign my review report  

( ) I would like to sign my review report  

English language and style

( ) Extensive editing of English language and style required  

(x) Moderate English changes required  

( ) English language and style are fine/minor spell check required  

( ) I don't feel qualified to judge about the English language and style  

Yes

Can be improved

Must be improved

Not applicable

Does the introduction provide sufficient background and include all relevant references?

(x)

( )

( )

( )

Is the research design appropriate?

(x)

( )

( )

( )

Are the methods adequately described?

(x)

( )

( )

( )

Are the results clearly presented?

(x)

( )

( )

( )

Are the conclusions supported by the results?

(x)

( )

( )

( )

Comments and Suggestions for Authors

Thank you for your attention to detail with the revised manuscript. The figures are more clear with the exception of Figure 5 and 7C which are difficult to read. 

Responses: The English spelling and grammar was improved and changed in the revised manuscript.

The figures were improved and submitted in better quality - 600 dpi. All figures are in the line of MDPIs' requirements.

This manuscript is a resubmission of an earlier submission. The following is a list of the peer review reports and author responses from that submission.

Round 1

Reviewer 1 Report

The study appears well performed but could be improved by attention to the following:

Major point

  1. L. 166-171

“The expression levels of selected TMEMs in tumor and normal tissue in patients with HNSCC were analyzed. Significant differences (p < 0.05) between expression levels of TMEMs were observed in 16 of 22 analyzed transcripts: ANO1, TMEM17, TMEM45B, TMEM48, TMEM97, TMEM140, TMEM156, TMEM158, TMEM173, TMEM206, RTP3, TMEM22, TMEM43, TMEM61, TMEM116 and TMEM213. For transcripts TMEM25, TMEM30B, TMEM45A, TMEM88, TMEM98 and TMEM176A no significant changes were observed (p > 0.05), Figure 1A.”

You need to clarify “up-regulated” or “down-regulated” in tumors compared to normal tissue for each gene, because some genes are difficult to be judged whether up-regulated or down-regulated.

  1. L. 249-250

“3.3. Only expression levels of ANO1, TMEM156, TMEM173 and TMEM213 transcripts influence on patients’ survival”

The analysis you performed doesn’t show any evidence that those gene expression influence on patients’ survival. You need to use the expression “are associate with” instead of “influence on”.

  1. L.268-338

This paragraph is too long. You need to summarize it.

4. The letters and numbers in Figure and supplementary Figure are too small. You need to make them easier to see.

Minor point

  1. Table 1 is hard to read. I think it will be better to put it into supplementary table.

  1. You need to indicate the Tables or Figures at first in each paragraph in the Result session, not the end.

  1. “Transcript(s)” should be replaced to “gene(s)”

  1. L. 23

“the gene set enrichment (GSEA) analysis”should be “the gene set enrichment analysis (GSEA)”

  1. L. 69-70

“It has been observed that many TMEM proteins are up- or downregulated in cancers.”

You need to indicate the source.

  1. L. 108

“age (<61 vs 61>)”

should be < 60 vs >= 61

  1. L. 312-318

“These changes were similar to these induced by modification in expression levels of BMI1, TAZ, YAP, KRAS, MYC, P53, EZH2, RB, TGF-β, PCGF2 and TBK1 expression, and changes in responses of cells to serum, genes up-regulated during late stages of differentiation of embryoid bodies, as well as similar to genes up-regulated in primary keratinocytes by expression of p50 (NFKB1) and p65 (RELA) components of NFKB, with NES between 1.642 and 2.217, p < 0.05 and FDR q < 0.25.”

It’s too long sentence to understanding. This sentence should be divided.

The gene name should be described in italics.

Author Response

Dear Editor and Reviewers,

Thank you very much for your revisions and suggestions. We have read them in detail and most of them were used by us to improve our manuscript. All of the changes were included in the current version of the draft and marked. The detailed responses point by point to the suggestions are included below and in the text.

We hope that the corrections made are in line with your suggestions and expectations, and are enough to publish our publication in the Cancers MDPI journal. 

We are glad that we could cooperate with you under the presented topic: diagnostic utility and biological role of TMEMs in HNSCC.

Best regards

Authors of manuscript

Review Report Form 1 

Open Review

(x) I would not like to sign my review report  

( ) I would like to sign my review report  

English language and style

( ) Extensive editing of English language and style required  

(x) Moderate English changes required  

( ) English language and style are fine/minor spell check required  

( ) I don't feel qualified to judge about the English language and style  

Yes

Can be improved

Must be improved

Not applicable

Does the introduction provide sufficient background and include all relevant references?

( )

(x)

( )

( )

Is the research design appropriate?

(x)

( )

( )

( )

Are the methods adequately described?

(x)

( )

( )

( )

Are the results clearly presented?

(x)

( )

( )

( )

Are the conclusions supported by the results?

(x)

( )

( )

( )

Comments and Suggestions for Authors

The study appears well performed but could be improved by attention to the following:

Major point

  1. L. 166-171

“The expression levels of selected TMEMs in tumor and normal tissue in patients with HNSCC were analyzed. Significant differences (p < 0.05) between expression levels of TMEMs were observed in 16 of 22 analyzed transcripts: ANO1, TMEM17, TMEM45B, TMEM48, TMEM97, TMEM140, TMEM156, TMEM158, TMEM173, TMEM206, RTP3, TMEM22, TMEM43, TMEM61, TMEM116 and TMEM213. For transcripts TMEM25, TMEM30B, TMEM45A, TMEM88, TMEM98and TMEM176A no significant changes were observed (p > 0.05), Figure 1A.”

You need to clarify “up-regulated” or “down-regulated” in tumors compared to normal tissue for each gene, because some genes are difficult to be judged whether up-regulated or down-regulated.

Response: The group of genes were divided into up- and downregulated.

  1. L. 249-250

“3.3. Only expression levels of ANO1, TMEM156, TMEM173 and TMEM213 transcripts influence on patients’ survival”

The analysis you performed doesn’t show any evidence that those gene expression influence on patients’ survival. You need to use the expression “are associate with” instead of “influence on”.

 Response: The phrase was corrected.

  1. L.268-338

This paragraph is too long. You need to summarize it.

 Response: The paragraph was shortened as possible.

  1. The letters and numbers in Figure and supplementary Figure are too small. You need to make them easier to see.

Response: Figures were improved as possible. 

Minor point

  1. Table 1 is hard to read. I think it will be better to put it into supplementary table.

Response: The table was placed in supplementary and the figure was created.

  1. You need to indicate the Tables or Figures at first in each paragraph in the Result session, not the end.

 Response: The figures were replaced according to the journal instructions after the first citation.

  1. “Transcript(s)” should be replaced to “gene(s)”

Response: The Transcript(s) were changed to “gene(s)”.

  1. L. 23

“the gene set enrichment (GSEA) analysis”should be “the gene set enrichment analysis (GSEA)”

 Response: Phrase was changed.

  1. L. 69-70

“It has been observed that many TMEM proteins are up- or downregulated in cancers.”

You need to indicate the source

Response: The literature references were added to the sentence: (https://pubmed.ncbi.nlm.nih.gov/30574087/ , https://pubmed.ncbi.nlm.nih.gov/31454669/ , https://pubmed.ncbi.nlm.nih.gov/33757462/)

  1. 108

“age (<61 vs 61>)”

should be < 60 vs >= 61

Response: It was corrected. 

  1. L. 312-318

“These changes were similar to these induced by modification in expression levels of BMI1, TAZ, YAP, KRAS, MYC, P53, EZH2, jRB, TGF-β, PCGF2 and TBK1 expression, and changes in responses of cells to serum, genes up-regulated during late stages of differentiation of embryoid bodies, as well as similar to genes up-regulated in primary keratinocytes by expression of p50 (NFKB1) and p65 (RELA) components of NFKB, with NES between 1.642 and 2.217, p < 0.05 and FDR q < 0.25.”

It’s too long sentence to understanding. This sentence should be divided.

Response: It was corrected and the whole paragraph was shortened.

The gene name should be described in italics.

Response: The gene names were corrected and written in italics.

Submission Date

07 June 2021

Date of this review

21 Jun 2021 04:06:52

Review Report Form 2 

Open Review

(x) I would not like to sign my review report  

( ) I would like to sign my review report  

English language and style

( ) Extensive editing of English language and style required  

( ) Moderate English changes required  

(x) English language and style are fine/minor spell check required  

( ) I don't feel qualified to judge about the English language and style  

Yes

Can be improved

Must be improved

Not applicable

Does the introduction provide sufficient background and include all relevant references?

( )

( )

(x)

( )

Is the research design appropriate?

( )

(x)

( )

( )

Are the methods adequately described?

( )

( )

(x)

( )

Are the results clearly presented?

( )

( )

(x)

( )

Are the conclusions supported by the results?

( )

(x)

( )

( )

Comments and Suggestions for Authors

The authors performed an interesting study evaluating the association between TMEM family genes with clinical, histologic, molecular, genomic, and immune features in head and neck cancer patients. This study represents a substantial undertaking and is of relevance to the head and neck cancer research community and cancer research community. Please consider the following points in your efforts to maximize the impact of the current manuscript:

  • Figures and Tables
    • The text is difficult or, at times, impossible to read. Response: tex was improved.
    • Table 1 should be condensed. It is too difficult to read and therefore interpret in its current form. Perhaps, a heatmap showing the TMEM expression data with clinical annotations above the plot would illustrate the point and the table could be put in a supplemental Excel file.  Response: The table was placed in supplementary and the figure was created.

  • Abstract
    • How did you select the 22 TMEMs to begin with? This would provide context for the reader. Response: it was explained.
    • The method for selecting TMEMs for further examination should be clarified. For example, sentence two of the Methods section of the Abstract reads, “Next, only changed TMEMs were examined depending on the clinical-pathological parameters.” What is meant by “changed TMEMs”? Changed relative to what? Response: it was explained.
    • When you say, “The pathway analysis using REACTOME and the GSEA was made,” recommend clarifying the categories of gene sets you are evaluating…such as immune, oncogenic… Response: It was corrected and clarified.
    • On line 30, the manuscript reads, “The expressions of TMEMs correlate transcripts involved in…” Does this refer to TMEM expression correlating with the given functional gene sets? Please clarify. Response: It was corrected.
    • On lines 30-32, it is not clear as written whether you are referring to pathway or gene level analyses. Response: It was corrected and clarified.
    • Lines 34-35 repeat ANO1 and TMEM173 on both lines. Is this meaning to point out a difference between HPV+ vs HPV- HNSCC? Clarifying this would be helpful.  Response: It was corrected and clarified.
    • The conclusion references the utility of these genes as biomarkers. What about the potential for targeting these pathways? 

Response: It is a good question. However, in our analysis we did not focus on the targeting of molecular pathways. In the "Discussion" section we included the using TMEMs as potential biomarkers for targeted therapy based on available literature.

  • Introduction
    • On lines 72-73, the manuscript states that TMEMs have prognostic relevance as biomarkers. Please provide references. Response: the references were added.
    • Why did you choose to focus on HNSCC? It is not clear from the Introduction. Response: It was clarified.
  • Materials and Methods
    • How did you choose this set of TMEMs to test? Response: It was clarified.
    • How did you process and normalize TCGA expression data? If you used downloaded data directly, which normalization method (e.g., RSEM, FPKM, RPKM) was used on the data you downloaded and did you process the data further? Response: It was clarified and described.
    • In section 2.2, please provide details regarding how differential expression analysis was performed between the tumor and normal tumor samples. Response: It was clarified and described.
    • Line 111: Angiolymphatic “dissection” should be “invasion”. Response: It was corrected.
    • Lines 112-114: Why were the TMEM expression data dichotomized and used in a stratified survival analysis as opposed to kept continuous and used in a Cox regression analysis? Why were the TMEM expression data dichotomized on the mean instead of the median, assuming a negative binomial distribution? determine levels of increased and decreased expression of the specified gene.

Response: The Reviewer 3 (below) suggested "If looking at the upper 5% and lower 5%, high and low expressers respectively, a greater difference may be seen." The correct cut off based on the median, mean or percentile are still artificial. It is difficult to choose a proper value and the best it will be used is the cut off based on the expression in normal samples. Unfortunately in the TCGA data there are only 44 normal samples and 522 cancer samples so, it is also difficult to determine levels of increased and decreased expression of the specified gene. The non-pathological expression level is difficult to assess.

Many authors use different types of divisions. We decided to use mean because it shows the shift direction of the gene expression in contrast to the using of the median or upper 5% and lower 5% of expression which cause the exclusion of many of the cases from the analyses.

However, we are aware that all of the methods used for estimation of the cut off have advantages and disadvantages and are based only on the assumptions.

  • Results
    • How did you use results from the correlation analysis among TMEM expression data to guide subsequent analyses? 

Response: this analysis helped us to draw conclusions about TMEMs functionality. It was additionally highlighted in the discussion 

  • On line 174, did TMEM97 have the highest number or highest proportion of significant correlations? For example, TMEM206 was correlated with 8 other TMEMs which is more than TMEM97. Response: TMEM97 is correlated with 12 other TMEMs (TMEM17, TMEM22, TMEM48, TMEM116, TMEM158, TMEM206, TMEM213, ANO1, TMEM45B, TMEM61, TMEM140 and TMEM17). It was additionaly explained in the text.
  • Table 1: Why do you present mean expression? I assume the data are normally distributed after a transformation that was performed in a processing step. Otherwise, medians might be preferable here.

Response: in most cases the mean is used because it shows the shift direction of the gene expression. It should be noted that all of the statistical tests were done in the proper way after checking the normality of distribution. Both tests: T-test and Manna-Whitney test were used depending on the normality of data. It is also difficult to present values as means and medians in one table or graph. We decided to use the mean of the value because it is the most common used in the comparison of two groups.

  • Figure 3: The numbers in each gene strata should be provided. This is important in determining the power to detect a difference between gene strata. This will also impact interpretation of the data. Response: additional data about “patients’ strata” was added to the supplementary material because of the keeping of clarity of graphs.
  • In section 3.4., why do you select ANO1, TMEM156, TMEM173, and TMEM213 for analysis? Response: it was clarified in the text.
  • Lines 376-388 / Figure 5C: Was this analysis performed using the Thorrson et al inferred immune cell abundances or did you use CIBERSORT or another deconvolution algorithm to infer immune cell abundance? If you used the Thorsson data, why not include the M2 macrophages abundances? Response: We used supporting data presented by Thorsson et al. as described in section 2.4. The missing analysis of M2 macrophages subpopulation was added.
  • Figure 6B survival curves: It would be helpful to include the number of patients in each gene strata, again for interpreting the power for detecting a difference between strata. Response: additional data was added to the graphs.

Submission Date

07 June 2021

Date of this review

21 Jun 2021 21:54:43

Review Report Form 

Open Review

(x) I would not like to sign my review report  

( ) I would like to sign my review report  

English language and style

(x) Extensive editing of English language and style required  

( ) Moderate English changes required  

( ) English language and style are fine/minor spell check required  

( ) I don't feel qualified to judge about the English language and style  

Yes

Can be improved

Must be improved

Not applicable

Does the introduction provide sufficient background and include all relevant references?

( )

( )

(x)

( )

Is the research design appropriate?

( )

( )

(x)

( )

Are the methods adequately described?

( )

( )

(x)

( )

Are the results clearly presented?

( )

( )

(x)

( )

Are the conclusions supported by the results?

( )

( )

(x)

( )

Comments and Suggestions for Authors

Title: The Landscape of Transmembrane Protein Family Members in Head and Neck Cancers: Their Biological Role and Diagnostic Utility

The authors aim to profile the TMEM landscape of HNSCC using in silico analyses. 522 HNSCC and 44 adjacent healthy tissue mRNA expression datasets were extracted from the TCGA database.  Comparative analysis between tumor and healthy were performed and found that 16 of 22 TMEMs were differentially expressed. Subsequently, the 16 TMEMs were subjected to extensive correlative analysis based on clinicopathological parameters. High and low expressers of each TMEM were determined using the mean mRNA expression as cut off, and a significant difference for overall survival (OS) was reported for ANO1, TMEM156, TMEM173 and TMEM213. In depth GSEA analysis was performed elucidating TMEM correlation with important processes such as metabolism, RNA processing, molecule trafficking and immunological response. In view of the REACTOME pathway enrichment analysis, immunological (C7) GSEA was performed. Elevated immune scores and statistically higher levels of immune cells fractions associated with increased TMEM156 and TMEM173 expression. Lastly, validation of results was performed using the Gene Expression Omnibus (GEO) data repository. Validation confirmed TMEM156 potential as a prognostic marker.

Overall, the manuscript is extremely poorly written to a point where it renders the whole unreadable. English grammar and spelling are frankly appalling. Multiple figures cannot be interpreted due to poor graphical quality or representation. Table 1 is uninterpretable.

The discussion is overly long and not justified by the descriptive analyses in the manuscript. Overinterpretations, short-circuited conclusions, etc.

These are major concerns based on the following points:

Major points:

  1. All data, with the exception of table 1 and figure 6, contain combinations of HPV + and HPV – patient expression. For example, GSEA analysis performed may have a bias towards a HPV class. Discrimination between these two populations would help stratify HNSCC populations based on TMEM expression.

- HPV location: HPV related cancers (HPV+) are mostly located in the oropharynx (predominantly at the tonsils and tongue base). For figure 2, this is important. https://www.ncbi.nlm.nih.gov/pmc/articles/PMC4846850/

- Example TMEM that is differentially expressed in HPV-/+

 - ANO1 amplification and overexpression associated with HPV negative distance metastasis. https://www.nature.com/articles/6605823

 - ANO1 is not amplified or overexpressed in patients that express HPV E6 and E7 https://onlinelibrary.wiley.com/doi/10.1002/ijc.24911

- HPV -/+ vary in immune infiltration https://www.ncbi.nlm.nih.gov/pmc/articles/PMC5070962/

Response: We are aware about the clinical and biological differences of HPV-positive HNSCC in comparison to HPV-negative. However, in the used data set from TCGA there was a lot of missing data about HPV-status. Only 73 patients were indicated as HPV-negative and 39 patients as HPV-positive. The rest of the patients have unknown status. There is no justification to consider patients with unknown HPV status to HPV-negative or HPV-positive. Because of missing data we decided to analyze the whole of HNSCC patients together. Our intention in this study was to describe the role of TMEMs in general, and indicate the potential role as biomarkers and try to describe the biological role in HNSCC. In the corrected version of discussion we added the description about HPV and changes in TMEMs. The suggested references are included in the current version of the manuscript. Moreover, we paid attention to our analysis and obtained results, which were made without division to HPV-negative and HPV-positive.

  1. Figure1: figures appear incorrect in order. Figure 1B and Figure 2 refer to tumor versus tumor, while Figure 1C is tumor versus healthy. Consider reordering.

Response: It is a good suggestion, but the changes in the figures' ordering will change the ordering set by us results. We decided to keep the figures ordered by us figures' order. In all descriptions we put attention to describing what is compared to what in each section.

  1. Figure 1B: 16 of the 22TMEMs were selected for HNSCC expression comparison between tumors. However, the remaining 6 TMEMs (removed as the P-value showed no difference between healthy and tumor) should have been compared between tumors. Possible sub class of tumor missed and to help further stratify HNSCC.

Response: It is a good suggestion, but we analyzed TMEMs in the general point of view in the case of HNSCC. Deeper analysis should be done but it influences the whole manuscript which in current form has a lot of information and is comprehensive. Adding the additional information will cause difficulties to understand our work. It should be emphasized that our publication is the first where so many TMEMs were described in HNSCC and for most of them there is no previously published works, which could be used to discuss the results obtained by us.

  1. Figure 1C: Authors compare HNSCC tumors versus healthy using ROC analysis. However, a figure missing is the comparison of healthy tissue against tumor based on specific location. For example, the larynx versus healthy larynx contributes to the AUC more than pharynx versus healthy pharynx.

Response: there were too few healthy and tumor samples after dividing on those locations to make such comparison possible.

  1. Figure 2 focuses on specific locations in HNSCC. Table1 then reverts to the HNSCC tumors as a whole. Figure 2 and table1, consider reordering.

Response: It was reordered.

  1. Figure 3: high and low expressing subgroups are separated for the DFS and OS curves. The threshold appears to be the mean. If looking at the upper 5% and lower 5%, high and low expressers respectively, a greater difference may be seen.

Response: It is a good suggestion but we decided to use mean because it shows the shift direction of the gene expression and in contrast to the using of "upper 5% and lower 5% of expression" it does not cause the exclusion of many of the cases from the analysed group.

  1. Figure 4B: throughout majority analysis (Figure1 correlation, sub-location, clinicopathological parameters, OS and GSEA), there is high correlation between TMEM156 and TMEM173. Is there known overlap in literature between these? How can there be no enriched pathways for TMEM213? 

Response: There is no literature evidence of association between TMEM156 and TMEM173. There were enriched pathways, however none of them were statistically significant (that is p value < 0.05 and FDR < 0.25).

  1. Figure 4B and 5A: different sample order on x axis.

Response: The order was corrected.

  1. Figure 5C: y-axis, what does Fraction mean in this case? Is this supposed to be the fraction of immune cells based on mRNA expression of immune specific mRNA? Percentage would clarify here.

Response: We used supporting data presented by Thorsson et al. as described in section 2.4. The authors presented the specified immune cells as the fractions so we decided to keep the current  value of these immune cells. Our decision is justified by clarity of the results. It  will be also helpful for readers if they want to check our results or make other, similar analyses based on the TMEMs level in HNSCC.

  1. Consistent spelling and sentence structure errors throughout the Introduction, results and figure legends.

Response: the errors were corrected in the current version.

- Missing title for figure 2 section.

Response: It was corrected.

- All figures are too small to read and interpret.

Response: The figures were improved and submitted in better quality - 600 dpi.

- Table 1: illegible. Table could be summarized with bar chart depicting mean and p-value. Optionally, place the rest in the supplemental data.

Response: The table was placed in supplementary and the figure was created.

Minor:

The manuscript is scattered with staccato and non-coherent sentences and statements.

Some examples:

Methods section:

“For further analysis only statistically significant were taken.”

This is neither a correct English sentence, nor does it state what the cut-off for statistical significance is, or what differences this is based on.

Response: incorrect sentences were improved and additional information was added.

In the discussion session:

“Contrarily, immune cells infiltrate tumors more in patients with higher TMEM156 and TMEM173. This is another fact supporting the hypothesis that those transcripts are involved in immune response against cancer”

Apart from the poor English Grammar, one descriptive finding does not render it a fact, let alone support a ‘hypothesis’ that is not a hypothesis. Further, transcripts in tumor cells do not control immune responses. If any, proteins do. I suspect the authors aim to refer to the TMEMs.

“A lot of immune cells possess antitumor or tumor-promoting properties”.

What purpose does a sentence like this serve?

Etc., etc., etc.

Response: incorrect sentences were improved and additional information was added. The shortcuts (simplifications) of sentences were removed and the sentences were clarified.

Submission Date

07 June 2021

Date of this review

09 Jul 2021 15:07:45

Reviewer 2 Report

The authors performed an interesting study evaluating the association between TMEM family genes with clinical, histologic, molecular, genomic, and immune features in head and neck cancer patients. This study represents a substantial undertaking and is of relevance to the head and neck cancer research community and cancer research community. Please consider the following points in your efforts to maximize the impact of the current manuscript:

  • Figures and Tables
    • The text is difficult or, at times, impossible to read.
    • Table 1 should be condensed. It is too difficult to read and therefore interpret in its current form. Perhaps, a heatmap showing the TMEM expression data with clinical annotations above the plot would illustrate the point and the table could be put in a supplemental Excel file.
  • Abstract
    • How did you select the 22 TMEMs to begin with? This would provide context for the reader.
    • The method for selecting TMEMs for further examination should be clarified. For example, sentence two of the Methods section of the Abstract reads, “Next, only changed TMEMs were examined depending on the clinical-pathological parameters.” What is meant by “changed TMEMs”? Changed relative to what?
    • When you say, “The pathway analysis using REACTOME and the GSEA was made,” recommend clarifying the categories of gene sets you are evaluating…such as immune, oncogenic…
    • On line 30, the manuscript reads, “The expressions of TMEMs correlate transcripts involved in…” Does this refer to TMEM expression correlating with the given functional gene sets? Please clarify.
    • On lines 30-32, it is not clear as written whether you are referring to pathway or gene level analyses.
    • Lines 34-35 repeat ANO1 and TMEM173 on both lines. Is this meaning to point out a difference between HPV+ vs HPV- HNSCC? Clarifying this would be helpful.
    • The conclusion references the utility of these genes as biomarkers. What about the potential for targeting these pathways?
  • Introduction
    • On lines 72-73, the manuscript states that TMEMs have prognostic relevance as biomarkers. Please provide references.
    • Why did you choose to focus on HNSCC? It is not clear from the Introduction.
  • Materials and Methods
    • How did you choose this set of TMEMs to test?
    • How did you process and normalize TCGA expression data? If you used downloaded data directly, which normalization method (e.g., RSEM, FPKM, RPKM) was used on the data you downloaded and did you process the data further?
    • In section 2.2, please provide details regarding how differential expression analysis was performed between the tumor and normal tumor samples.
    • Line 111: Angiolymphatic “dissection” should be “invasion”.
    • Lines 112-114: Why were the TMEM expression data dichotomized and used in a stratified survival analysis as opposed to kept continuous and used in a Cox regression analysis? Why were the TMEM expression data dichotomized on the mean instead of the median, assuming a negative binomial distribution?
  • Results
    • How did you use results from the correlation analysis among TMEM expression data to guide subsequent analyses?
    • On line 174, did TMEM97 have the highest number or highest proportion of significant correlations? For example, TMEM206 was correlated with 8 other TMEMs which is more than TMEM97.
    • Table 1: Why do you present mean expression? I assume the data are normally distributed after a transformation that was performed in a processing step. Otherwise, medians might be preferable here.
    • Figure 3: The numbers in each gene strata should be provided. This is important in determining the power to detect a difference between gene strata. This will also impact interpretation of the data.
    • In section 3.4., why do you select ANO1, TMEM156, TMEM173, and TMEM213 for analysis?
    • Lines 376-388 / Figure 5C: Was this analysis performed using the Thorrson et al inferred immune cell abundances or did you use CIBERSORT or another deconvolution algorithm to infer immune cell abundance? If you used the Thorsson data, why not include the M2 macrophages abundances?
    • Figure 6B survival curves: It would be helpful to include the number of patients in each gene strata, again for interpreting the power for detecting a difference between strata.

Author Response

(The authors gave the same response as above.)

Reviewer 3 Report

Title: The Landscape of Transmembrane Protein Family Members in Head and Neck Cancers: Their Biological Role and Diagnostic Utility

The authors aim to profile the TMEM landscape of HNSCC using in silico analyses. 522 HNSCC and 44 adjacent healthy tissue mRNA expression datasets were extracted from the TCGA database.  Comparative analysis between tumor and healthy were performed and found that 16 of 22 TMEMs were differentially expressed. Subsequently, the 16 TMEMs were subjected to extensive correlative analysis based on clinicopathological parameters. High and low expressers of each TMEM were determined using the mean mRNA expression as cut off, and a significant difference for overall survival (OS) was reported for ANO1, TMEM156, TMEM173 and TMEM213. In depth GSEA analysis was performed elucidating TMEM correlation with important processes such as metabolism, RNA processing, molecule trafficking and immunological response. In view of the REACTOME pathway enrichment analysis, immunological (C7) GSEA was performed. Elevated immune scores and statistically higher levels of immune cells fractions associated with increased TMEM156 and TMEM173 expression. Lastly, validation of results was performed using the Gene Expression Omnibus (GEO) data repository. Validation confirmed TMEM156 potential as a prognostic marker.

Overall, the manuscript is extremely poorly written to a point where it renders the whole unreadable. English grammar and spelling are frankly appalling. Multiple figures cannot be interpreted due to poor graphical quality or representation. Table 1 is uninterpretable.

The discussion is overly long and not justified by the descriptive analyses in the manuscript. Overinterpretations, short-circuited conclusions, etc.

These are major concerns based on the following points:

Major points:

  1. All data, with the exception of table 1 and figure 6, contain combinations of HPV + and HPV – patient expression. For example, GSEA analysis performed may have a bias towards a HPV class. Discrimination between these two populations would help stratify HNSCC populations based on TMEM expression.

- HPV location: HPV related cancers (HPV+) are mostly located in the oropharynx (predominantly at the tonsils and tongue base). For figure 2, this is important. https://www.ncbi.nlm.nih.gov/pmc/articles/PMC4846850/

- Example TMEM that is differentially expressed in HPV-/+
 - ANO1 amplification and overexpression associated with HPV negative distance metastasis. https://www.nature.com/articles/6605823
 - ANO1 is not amplified or overexpressed in patients that express HPV E6 and E7 https://onlinelibrary.wiley.com/doi/10.1002/ijc.24911

- HPV -/+ vary in immune infiltration https://www.ncbi.nlm.nih.gov/pmc/articles/PMC5070962/

  1. Figure1: figures appear incorrect in order. Figure 1B and Figure 2 refer to tumor versus tumor, while Figure 1C is tumor versus healthy. Consider reordering.
  2. Figure 1B: 16 of the 22TMEMs were selected for HNSCC expression comparison between tumors. However, the remaining 6 TMEMs (removed as the P-value showed no difference between healthy and tumor) should have been compared between tumors. Possible sub class of tumor missed and to help further stratify HNSCC.
  3. Figure 1C: Authors compare HNSCC tumors versus healthy using ROC analysis. However, a figure missing is the comparison of healthy tissue against tumor based on specific location. For example, the larynx versus healthy larynx contributes to the AUC more than pharynx versus healthy pharynx.
  4. Figure 2 focuses on specific locations in HNSCC. Table1 then reverts to the HNSCC tumors as a whole. Figure 2 and table1, consider reordering.
  5. Figure 3: high and low expressing subgroups are separated for the DFS and OS curves. The threshold appears to be the mean. If looking at the upper 5% and lower 5%, high and low expressers respectively, a greater difference may be seen.
  6. Figure 4B: throughout majority analysis (Figure1 correlation, sub-location, clinicopathological parameters, OS and GSEA), there is high correlation between TMEM156 and TMEM173. Is there known overlap in literature between these? How can there be no enriched pathways for TMEM213?
  7. Figure 4B and 5A: different sample order on x axis.
  8. Figure 5C: y-axis, what does Fraction mean in this case? Is this supposed to be the fraction of immune cells based on mRNA expression of immune specific mRNA? Percentage would clarify here.
  9. Consistent spelling and sentence structure errors throughout the Introduction, results and figure legends.

- Missing title for figure 2 section.

- All figures are too small to read and interpret.

- Table 1: illegible. Table could be summarized with bar chart depicting mean and p-value. Optionally, place the rest in the supplemental data.

Minor:

The manuscript is scattered with staccato and non-coherent sentences and statements.

Some examples:

Methods section:

“For further analysis only statistically significant were taken.”

This is neither a correct English sentence, nor does it state what the cut-off for statistical significance is, or what differences this is based on.

In the discussion session:

“Contrarily, immune cells infiltrate tumors more in patients with higher TMEM156 and TMEM173. This is another fact supporting the hypothesis that those transcripts are involved in immune response against cancer”

Apart from the poor English Grammar, one descriptive finding does not render it a fact, let alone support a ‘hypothesis’ that is not a hypothesis. Further, transcripts in tumor cells do not control immune responses. If any, proteins do. I suspect the authors aim to refer to the TMEMs.

“A lot of immune cells possess antitumor or tumor-promoting properties”.

What purpose does a sentence like this serve?

Etc., etc., etc.

Author Response

(The authors gave the same response as above.)
